# A Novel LHX6 Reporter Cell Line for Tracking Human iPSC-Derived Cortical Interneurons

**DOI:** 10.3390/cells11050853

**Published:** 2022-03-01

**Authors:** Maria Cruz-Santos, Lucia Fernandez Cardo, Meng Li

**Affiliations:** 1Neuroscience and Mental Health Research Institute, School of Medicine, Cardiff University, Cardiff CF24 4HQ, UK; cruzsantosmc@cardiff.ac.uk (M.C.-S.); cardolf@cardiff.ac.uk (L.F.C.); 2School of Biosciences, Cardiff University, Cardiff CF10 3AX, UK

**Keywords:** CRISPR/Cas9, genome editing, human pluripotent stem cell, in vitro differentiation, GABA, interneuron, LHX6

## Abstract

GABAergic interneurons control the neural circuitry and network activity in the brain. The dysfunction of cortical interneurons, especially those derived from the medial ganglionic eminence, contributes to neurological disease states. Pluripotent stem cell-derived interneurons provide a powerful tool for understanding the etiology of neuropsychiatric disorders, as well as having the potential to be used as medicine in cell therapy for neurological conditions such as epilepsy. Although large numbers of interneuron progenitors can be readily induced in vitro, the generation of defined interneuron subtypes remains inefficient. Using CRISPR/Cas9-assisted homologous recombination in hPSCs, we inserted the coding sequence of mEmerald and mCherry fluorescence protein, respectively, downstream that of the *LHX6*, a gene required for, and a marker of medial ganglionic eminence (MGE)-derived cortical interneurons. Upon differentiation of the LHX6-mEmerald and LHX6-mCherry hPSCs towards the MGE fate, both reporters exhibited restricted expression in LHX6^+^ MGE derivatives of hPSCs. Moreover, the reporter expression responded to changes of interneuron inductive cues. Thus, the LHX6-reporter lines represent a valuable tool to identify molecules controlling human interneuron development and design better interneuron differentiation protocols as well as for studying risk genes associated with interneuronopathies.

## 1. Introduction

Cortical interneurons are gamma-aminobutyric acid-containing (GABAergic) inhibitory neurons that connect locally in the neocortex. They provide inhibitory inputs that shape the responses of pyramidal cells and prevent runaway excitation [1,2,3,4]. Cortical interneurons regulate the timing and synchronization of population rhythms expressed as cortical neuronal oscillations, thereby modulating cortical network activity with high spatiotemporal control [1,5,6]. Increasing evidence suggests that disruption of interneuron function underlies neuropsychiatric conditions such autism, schizophrenia and epilepsy [2,7,8,9,10,11,12].

Cortical interneurons are a heterogeneous cell population consisting of three major subtypes based on their expression of parvalbumin (PV), somatostatin (SST) and ionotropic serotonin receptor 5HT3a (5HT3aR), respectively [4,13,14,15]. The PV and SST interneurons originate from the Nkx2.1^+^ neural progenitors of the medial ganglionic eminence (MGE), while the 5HT3aR interneurons are derived from the dorsal caudal ganglionic eminence (CGE). CGE neural progenitors express Coup tfii and Prox1, but not Nkx2.1. The PV and SST neurons are the most numerous subtypes in the cortex [16]. They are direct progeny of nascent neurons expressing the LIM homeodomain transcription factor Lhx6 that has restricted expression in the MGE [15,17,18]. Apart from serving as a lineage-specific marker of SST and PV cortical interneurons, LHX6 is required for the generation of SST and PV neurons as well as their tangential migration from the MGE to the cortex [14,19,20,21]. A recent single cell transcriptomics study revealed that humans and mice share transcriptional programs for interneuron development [22], providing further support for using LHX6 as a lineage marker for human MGE-derived cortical interneurons. Pluripotent stem cells (PSCs) can give rise to all cell types of the body when exposed to the right inductive cues at appropriate time windows during in vitro differentiation [23], thus providing a powerful tool for modelling human development and diseases. Several methods have been reported to generate GABAergic neurons from human PSCs (hPSCs) [24,25,26,27,28,29]. These pioneering works revealed that, while MGE-like progenitors (NKX2.1^+^) can be abundantly produced, the generation of the LHX6 lineage interneurons, i.e., those that express PV or SST, remains inefficient. The current state-of-the-art reflects our limited knowledge about transcription factor regulation and signaling pathways that control MGE development and interneuron fate specifications despite recent significant advances [22,30,31].

PSCs engineered with a lineage or cell type specific reporters have proven to be valuable tools for identifying novel determinants of a defined cell fate, or purifying cell types of interest for a number of downstream applications [24,32,33,34]. Although an NKX2.1-GFP hESC reporter line for monitoring hESC-derived MGE neural progenitors and a VGAT-GFP hESC line are available [34,35], these reporter lines are not specific to PV and SST interneurons because MGE also give rise to globus pallidus local projection neurons and cholinergic basal forebrain neurons, while VGAT marks all GABAergic neurons [15,17,18]. Moreover, Nkx2.1 expression is down regulated in mature cortical interneurons [36]. To facilitate the identification of novel regulators of cortical interneuron differentiation and to improve in vitro paradigms for their production, we generated LHX6-mEmerald and LHX6-mCherry reporter cell lines for tracking hPSCs-derived cortical interneurons by CRISPR/Cas9-assisted gene targeting. These LHX6 reporter lines faithfully express mCherry and mEmerald in LHX6^+^ neurons of hPSC-derived MGE progeny.

## 2. Materials and Methods

### 2.1. CRISPR/Cas9-Assisted Gene Targeting

Three gRNAs were designed using the online tool (https://zlab.bio/guide-design-resources, accessed on 22 May 2016) to target the 3′ UTR of the human *LHX6* gene. Using a multiplex CRISPR/Cas9 assembly system [37], the gRNAs were first cloned into individual vectors before Golden Gate assembly into the destination vector, which also contains a Cas9 expression unit. The resultant plasmid was named Cas9/3xsgRNAs. The LHX6 targeting vector contains either a p2A-mEmerald or a p2A-mCherry sequence followed by a LoxP-flanked neomycin drug selection cassette between the 5′ and 3′ homology arms, corresponding to the 500 bp sequence upstream of the first LHX6 stop codon and 3′ UTR, respectively. The two plasmids were co-nucleofected into H7 hESCs (mCherry) and Kolf2 hiPSCs (mEmerald, https://hpscreg.eu/cell-line, accessed on 22 May 2016) at a ratio of 1 Cas9/3xsgRNAS: 3 targeting vector. Individual G418-resistant colonies were picked, expanded and genotyped by PCR. Targeted clones identified by PCR were further verified by Sanger sequencing. Selected targeted clones were then transiently transfected with a Cre-expressing plasmid and individual clones isolated again. The loss of the drug selection cassette was confirmed by PCR and G418 resistance test. The cell lines used in this study were two homozygous mCherry lines and one genotype each of the mEmerald lines. All lines were genotype verified by microarray-based CNV analysis (Illumina, San Diego, CA, USA).

### 2.2. HESC Culture and Neuronal Differentiation

Two independent clones of the LHX6-mEmerald and LHX6-mCherry reporter lines were used in this study. The parental (isogenic) H7 and KOLF2 lines were used as negative control where relevant. The two clones of mCherry and mEmerald lines behaved indistinguishably in terms of respective fluorescence intensity and pattern and were therefore referred to in short as LHM for the mCherry and LHE for the mEmerald lines. 

Routine hPSC culture and MGE differentiation followed procedures described previously [29]. All hPSCs were cultured on Matrigel-coated plastics in E8 media (Thermo Fisher, Inchinnan, UK). The media was changed daily and the cells passaged mechanically with 0.02% EDTA at 80% confluence. For MGE differentiation, cells from two 80% confluent wells of a 6-well plate were plated onto a 12-well plate previously coated with reduced growth factor Matrigel (VWR) in E8 media (day 0) and changed to N2B27 the next day. The cells were induced to neuroectoderm fate by LDN-193189, SB-431542 and XAV-939 from day 1 to 10, followed by SHH and SHH agonist purmorphamine (PM) induction of ventralization from day 11 to 20. To promote terminal differentiation and cell survival, the cultures were treated with BDNF from day 25 until analysis. Cortical differentiation follows the same procedure without XAV, SHH and PM.

### 2.3. Immunohistochemistry

The cultures were fixed with 3.7% PFA for 15–20 min at 4 °C. For nuclear antigen detection, an additional fixation with methanol gradient was performed, which included 5 min each in 33% and 66% methanol at room temperature, followed by 100% methanol for 20 min at −20 °C. The cultures were then returned to PBS-T (0.3% Triton-X-100 solution in PBS) via inverse gradient and were then permeabilized during three 10-minute washes and then blocked in PBS-T containing 1% BSA and 3% donkey serum. The cells were incubated with primary antibodies in blocking solution overnight at 4 °C. Following three PBS-T washes, Alexa-Fluor secondary antibodies (Thermo Fisher Scientific) were added at 1:1000 in blocking solution for 1 h at ambient temperature in the dark. Three PBS-T washes were then performed that included once with DAPI at 1:1000 (Thermo Fisher Scientific). Images were taken on a Zeiss LSM710 confocal microscope from at least 5 randomly selected fields/samples and staining quantification was acquired manually in ImageJ1.51 (imagej.net; accessed on 22 May 2016).

### 2.4. Flow Cytometry

The cultures were dissociated into single cell suspension using Accutase at 37 °C for 5–10 min, depending on the stage of differentiation. The cells were washed with cold PBS, followed by centrifugation, followed by resuspension in 200 μL of cold PBS containing DAPI (1:6000). The data for mCherry+ cells were acquired using a Fortessa analyzer. Parental H7 cells at the same stage of differentiation were used at each time-point as negative controls. DAPI staining was used to discard dead cells from the analysis. 

### 2.5. Statistical Analyses

All data were collected from at least three independent experiments and are presented as mean ± SEM. The data were tested for normality with the Shapiro–Wilk test and for equal variance with the Levene test before performing statistical analyses by unpaired *t*-test or non-parametric alternatives, as stated in the figure legends where relevant. All statistical tests were performed in SPSS (IBM, Armonk, NY, USA).

## 3. Results

### 3.1. Targeting mCherry and mEmerald into the LHX6 Locus of hPSCs

A targeting vector was designed to introduce a p2A-mCherry or p2A-mEmerald and a floxed PGKNeo cassette to the 3′ UTR of the endogenous *LHX6* locus (Figure 1A). The human *LHX6* gene encodes nine protein coding transcripts, which use either of two stop codons in exon 10. In order to link mCherry and mEmerald expression to all of these transcripts, the 500 bp genomic sequence immediately upstream of the first stop codon was chosen as the 5′ homology arm, while the 3′ homology arm corresponded to 500 bp of the *LHX6* 3′ UTR immediately downstream of the second stop codon. Following electroporation, drug selection and clonal amplification, genomic PCR identified 10 heterozygous and two homozygous mEmerald knocking clones and two each heterozygous and homozygous mCherry targeted clones out of 48 and 37 colonies, respectively (Figure 1B,C and Appendix A). Further Sanger DNA sequencing confirmed in-frame integration of the p2A-mEmerald and p2A-mCherry sequence downstream of the last coding codon of *LHX6* (Figure 1D and Appendix A). The neo resistant cassette was then removed by transient Cre expression in order to prevent potential unpredictable effects on reporter expression [38,39,40]. PKG-Neo excision was verified by genomic PCR after clonal isolation (Figure 1C and Appendix A). All Neo-deleted colonies retained characteristic morphology (Figure 2A) and two independent clones from each reporter line were further tested for pluripotency marker expression and chromosome count. Almost all cells were OCT4^+^ and NANOG^+^, while approximately 82% of the nuclei contained 46 chromosomes (Figure 2B,C). The genomic integrity of these clones was also verified by array-based CNV analysis (Appendix A). Moreover, following random differentiation of these cells in serum-supplemented DMEM media for 15 days in the absence of specific inductive molecules, both cultures contained cells expressing Brachyury and Eomesodermin (Eomes), two T-Box transcription factors that control pluripotency exit and mesoderm and definitive endoderm programs (Figure 2D) [41]. These clones were used in subsequent studies and referred to thereafter as LHM for the mCherry and LHE for the mEmerald lines, respectively.

### 3.2. mCherry and mEmerald Expression Faithfully Mirror That of LHX6 during hPSC Interneuron Differentiation

We first tested the MGE differentiation capacity of the LHM and LHE reporter lines compared to H7 hESCs using a published protocol (Figure 3A) [29]. Following ventralization with SHH and purmorphamine, neural progenitors derived from both reporter lines acquired MGE characteristics at day 21, as demonstrated by a highly enriched population of cells expressing FOXG1 and NKX2.1, which marks forebrain and MGE, respectively (Figure 3B). The proportion of NKX2.1^+^ and FOXG1^+^ cells in the LHM and LHE cultures were similar to that in the parental control H7 cultures, while none of these MGE induced cultures contained cells positive for PAX6, a marker for dorsal telencephalic (cortical) neural progenitors (Figure 3B,C and Appendix A). These observations suggest that LHM and LHE reporter hPSCs can be efficiently induced towards the MGE fate. 

We then verified the fidelity of the reporter expression by following the temporal progression of mCherry+ and mEmerald+ cells during a 45-day interneuron differentiation time window by flow cytometry and fluorescence microscopy. At day 25, around 10% of the cells in the LHM cultures were detected as mCherry+ by flow cytometry. The proportion of mCherry+ cells rose to just over 21% at day 35 before reducing at day 45 (Figure 3D). The double immunostaining for LHX6 and mCherry of a day 40 culture revealed a near complete overlap between the two staining (Figure 3E and Appendix A). 

As a second control, the LHM and LHE cells were also differentiated under a cortical protocol to investigate potential reporter expression outside the MGE lineage (Figure 3B–D,F). The majority of the cells in these cortical cultures stained positive for PAX6, while NKX2.1^+^ cells were rare (Figure 3B,C), confirming efficient cortical fate induction in both the LHM and LHE cultures. However, few cells were present in the mCherry+ fraction in LHM-derived cortical differentiation cultures at any of the differentiation time points analyzed by flow cytometry (Figure 3D), and no mEmerald+ cells were detected in the LHE-derived cortical cultures by fluorescence microscopy (Figure 3F and Appendix A). 

In contrast, epifluorescence microscopy examination of MGE-induced LHE cultures from day 5 detected evident mEmerald+ cells from day 25 onwards (Figure 3F and Appendix A). The mEmerald signal increased gradually as cultures progressed from day 25 to 45, while neuronal morphology and neuronal processes became more prominent (Figure 3G). However, mCherry+ cells could not be visualized directly by fluorescence microscopy.

To provide additional evidence that the expressions of the mCherry and mEmerald reporter are restricted to the LHX6 branch cortical interneurons, we let the MGE-induced LHM and LHE cultures to differentiate further till day 60, when cells expressing interneuron subtype markers such as SST began to be detected. We found that SST^+^ cells were also mCherry+ or mEmerald+ (Figure 3H,I and Appendix A). Together, these findings demonstrate that the expression of the two LHX6 knock-in reporters is restricted to hPSC-derived MGE-like neurons. 

### 3.3. Production of mEmerald+ and mCherry+ Cells during PSC Differentiation Respond to External Cues

An anticipated application of the LHX6 reporter hPSC lines is as a tool to identify novel factors that regulate cortical interneuron induction and/or differentiation. Having ascertained that the mEmerald and mCherry faithfully mirror that of LHX6 expression during PSC differentiation, we next evaluated the responsiveness of the reporters to changes of inductive paradigm. In the first test, we shifted the exposure of ventralization cues from day 10 to 20 to an earlier time window of day 3 to 10 (SHIFTED condition) and examined its effect on MGE induction and reporter expression using the LHX6-mEmerald line (Figure 4). Compared to the ‘standard’ (STD) condition, adding SHH and purmorphamine early during PSCs transition from pluripotent state to neural fate had a negative impact on MGE fate induction, as indicated by the reduced proportion of NKX2.1+ neural progenitors at day 21 (Figure 4B,C). Fluorescence microscopy examination detected few mEmerald+ cells in the SHIFTED cultures compared to the abundant presence of mEmerald+ neurons in the STD control cultures at days 25, 35 and 45 (Figure 4D). Double immunostaining for SST and the pan GABAergic marker GAD67 at day 65 revealed a reduction of SST+ cells, while the total GABAergic content was comparable between the two culture conditions (Figure 4E,F). 

Since LHX6 is expressed in nascent postmitotic cortical interneurons [21,42], we tested reporter response to the modulation of cell division. As shown in Figure 3D, we observed a decrease in the proportion of mCherry+ cells from day 35 to day 45. Within this time window, the cell density continued to increase; we therefore postulated that the decrease in the mCherry+ cell proportion may be due to the continued proliferation of LHX6^-^ neural progenitors, leading to a ‘dilution’ of LHX6-mCherry+ cells in the cultures. To test this hypothesis, we treated day 35 cultures for 2 h with demecolcine, a drug that inhibits mitosis by inhibiting spindle formation during cell division (Figure 4G). Under this condition, we observed a further rise in the mCherry+ population till day 40 and the level remained relatively stable thereafter. This experiment indicates that an increase in the cell cycle exit of MGE progenitors can be readily detected by changes in LHX6-reporter-expressing cells. Together, the above experiments provide proof of principle that the LHX6 reporter lines could be used to identify factors that regulate, either positively or negatively, cortical interneuron induction during hPSC differentiation.

## 4. Discussion

Using CRISPR/Cas9-assisted gene targeting, we successfully generated knock-in lines in the *LHX6* locus for tracking cortical interneurons derived from hPSCs. The mCherry and mEmerald reporters are faithfully expressed in LHX6-expressing neurons amongst in vitro differentiated MGE derivatives. No leakage of reporter expression was detected in undifferentiated hPSCs and neuronal cultures directed towards cortical fates in both reporter lines. 

LHX6 is a transcription factor critically required for interneuron differentiation. Hence, a key consideration of our targeting strategy concerns safeguarding LHX6 gene expression and protein integrity. By inserting mEmerald and mCherry, respectively, into the 3′ UTR and retaining all LHX6 coding sequences, the *LHX6* gene remains intact, thus allowing the generation of homozygous knock-in lines without comprising the MGE differentiation capacity. 

Using the same targeting strategy and vector design, our study demonstrates that mEmerald is superior to mCherry as a fluorescent protein reporter, at least in the *LHX6* locus. The Emerald signal could be readily visualized by conventional epifluorescence microscopy in heterozygous LHX6-mEmerald lines before and after the removal of the drug selection cassette. However, mCherry expression could only be detected by flow cytometry and only in homozygous knock-in lines after the selection cassette was excised.

Currently, the robust generation of cortical interneurons from hPSCs remains a challenge. Although the induction of NKX2.1^+^ MGE progenitors can be achieved reliably and highly efficiently [24,25,26,27,28,29], only a proportion of these progenitors progress to LHX6^+^ postmitotic interneurons. Moreover, the yield of defined interneuron subtypes (SST^+^ or PV^+^) remains very low [24,25,26,27,28,29], or the phenotype unstable [43]. This is largely due to our limited understanding of the molecular machinery controlling interneuron diversity and sub-type specification, which is further complicated by the protracted acquisition of interneuron subtype identity both during normal development and hPSC differentiation [25,44,45]. The faithful expression of mEmerald and mCherry in the hPSC-derived LHX6 branch of MGE derivatives highlights the value of these reporter lines for the future identification of novel molecules and signaling pathways that play a role in cortical interneuron specification and/or differentiation. Of particular potential significance, the direct visualization of mEmerald+ interneurons, including neuronal processes, offers opportunities for image-based high content analysis for screening drugs that regulate cortical interneuron maturation, survival and function.

In summary, this work established a valuable tool to study human cortical interneuron development. It is anticipated that these reporter cells could also be used for investigations into the etiology of neuropsychiatric diseases in combination with CRISPR/CAS9 genome editing of associated risk genes.

## Figures and Tables

**Figure 1 cells-11-00853-f001:**
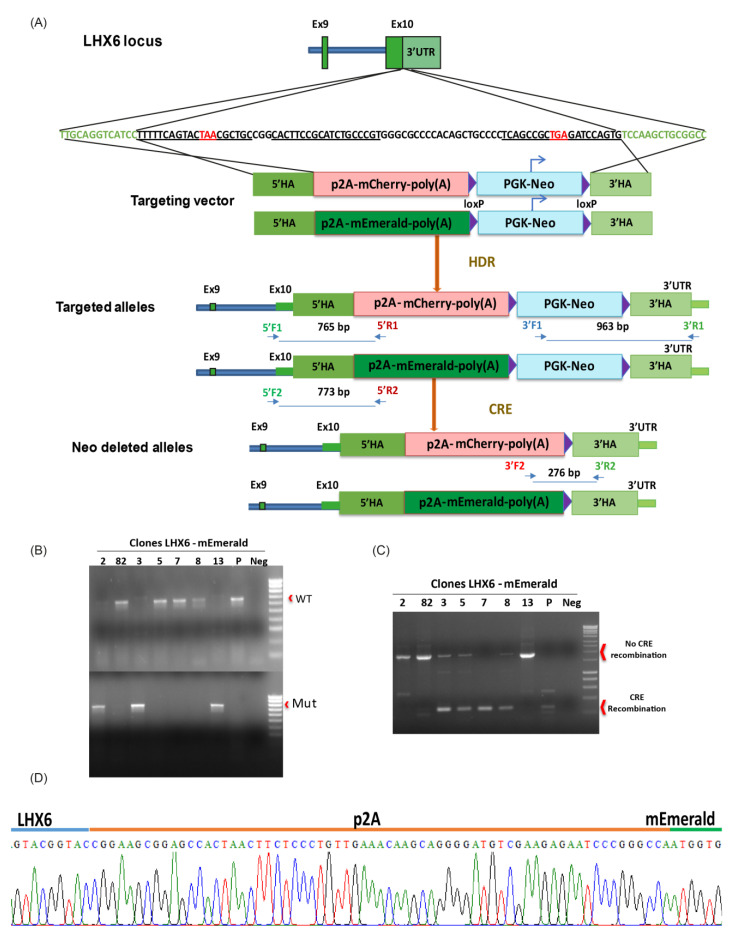
Generation of the LHX6 reporter lines. (**A**) Schematic overview of LHX6 targeting strategy. The darker green and pale green bars indicate coding parts of exons and 3′ UTR, respectively. The sequence underlined denotes the three gRNAs, while the stop codons are shown in red. The narrower green bars in the targeted allele represent part of the exon or 3′ UTR outside the homology. The arrowed lines under ‘targeted allele’ indicate the positions and predicted size of PCR amplicon of the 5′ and 3′ primer pairs. (**B**) Genomic PCR detection of the WT and targeted allele using the 5′ primer pairs. (**C**) PCR verification of Neo cassette removal. (**D**) An example of sequence read of 5′ PCR product in a targeted clone confirming in frame integration of p2A-mEmerald immediately downstream of the last coding codon of LHX6.

**Figure 2 cells-11-00853-f002:**
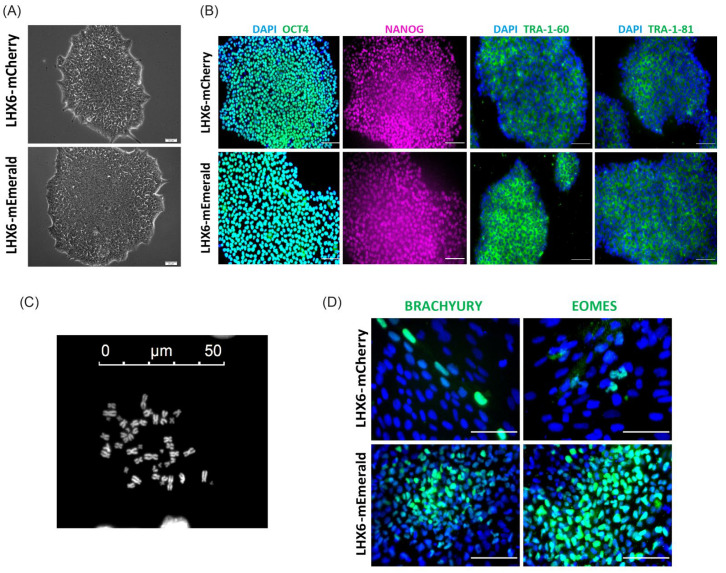
Reporter lines preserve pluripotent characteristics. (**A**) Phase contrast image of LHX6-mCherry and LHX6-mEmerald colonies showing characteristic hPSC morphology. (**B**) Double antibody staining for OCT4 and NANOG, and single staining for TRA 1–60 and TRA 1–81, respectively. (**C**) An example image of chromosome spread. (**D**) Immunostaining for Brachyury Y^+^ and EOMES in 15-day random differentiated LHE and LHM cultures. Scale bars in (**A**,**C**,**D**): 50 µ; (**B**): 100 µ.

**Figure 3 cells-11-00853-f003:**
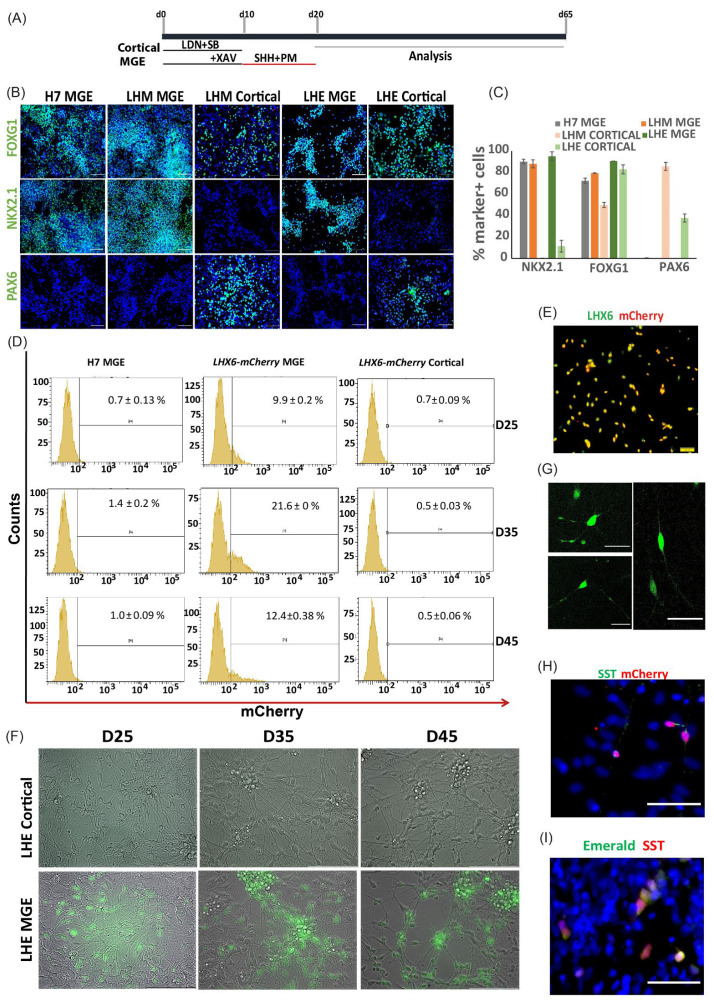
mCherry and mEmerald expression is restricted to MGE derivatives. (**A**) Schematic illustration of hPSC interneuron and cortical differentiation protocol. (**B**) Day 21 LHM. LHE and H7 differentiation cultures were immunostained for neural progenitor markers representing pan-forebrain (FOXG1), MGE (NKX2.1) and developing cortex (PAX6) with DAPI counter stain (blue). (**C**) Quantification of antibody staining exemplified in B. Bar graphs represents mean ± SEM of three independent differentiation runs. (**D**) mCherry signal detected by flow cytometry during MGE differentiation of LHM cells; virtually no signal was detected in H7 MGE- or LHM-cortical differentiation. Data represent mean ± SEM of three independent experiments. (**E**) Double immunostaining for LHX6 and mCherry in LHM MGE-differentiated cultures showing nearly complete colocalization of both proteins. (**F**) Epifluorescence microscopy of MGE and cortical differentiated LHE cells showing specific mEmerald signal in MGE cultures only. (**G**) Confocal microscopy of day 50 live cultures of LHE MGE differentiation reveals neuronal process of cells of different morphology. (**H**) Double immunostaining of SST and mCherry. (**I**) Double immunostaining of GFP and SST. Scale bars (**B**): 100µ; (**E**,**G**–**I**): 50 µ.

**Figure 4 cells-11-00853-f004:**
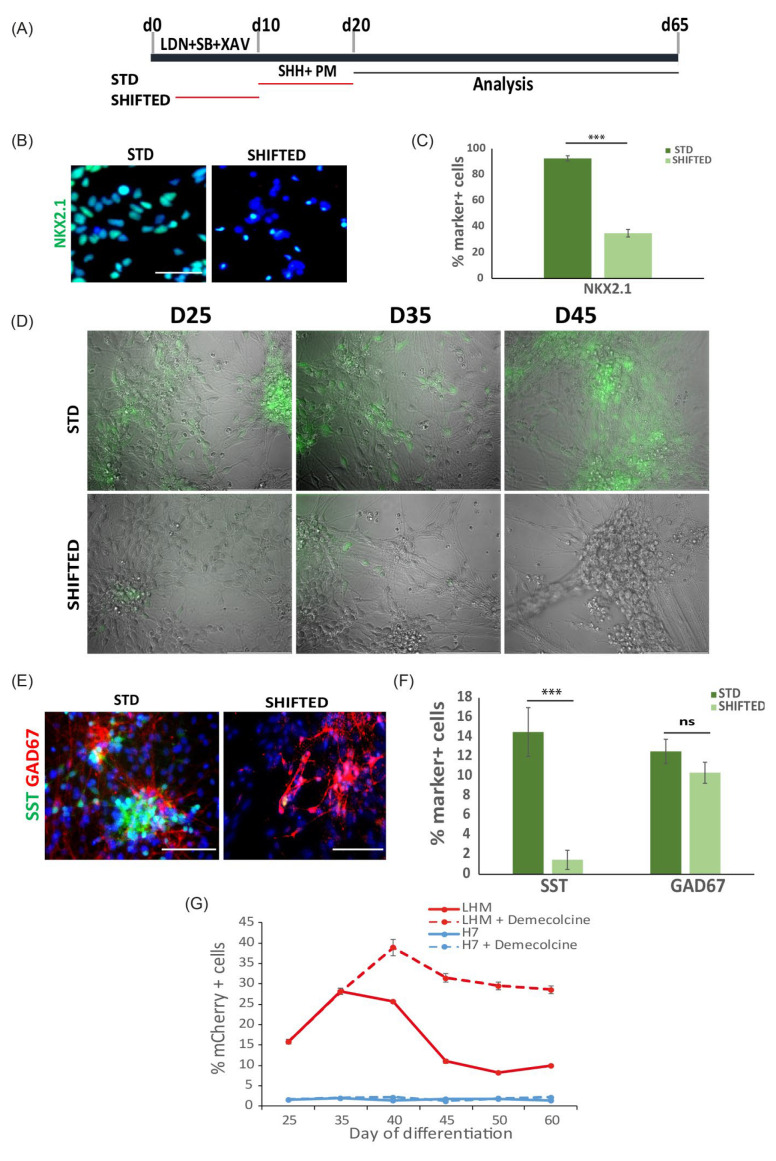
Reporter expression response to inductive signals and other cues. (**A**) Illustration of standard and inductive cue ‘SHIFTED’ MGE differentiation paradigms. (**B**) Day 21 cultures stained for NKX2.1 with DAPI counter stain. (**C**) Quantification of NKX2.1^+^ cells. Data represent the mean ± SEM from three independent experiments, *** *p* < 0.001 two tailed unpaired *t*-test, equal variances and normal distribution tested. (**D**) Epifluorescence microscopy during MGE differentiation of LHE cells. (**E**) Double immunostaining for SST and GAD67 in STD and shifted conditions. (**F**) Quantification of SST^+^ and GAD67^+^ cells in E. *n* = 3; ns, *p* > 0.05, *** *p* < 0.001 two tailed unpaired *t*-test, equal variances and normal distribution tested. (**G**) Flow cytometry measurement of mCherry^+^ cells during 60-day differentiation under conditions indicated. Data are presented as mean ± SEM of three independent experiments. Scale bar in (**B**,**E**): 50 μm.

## Data Availability

The raw data that support these findings are available on request to the corresponding authors.

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
