# Peer review of "A Novel LHX6 Reporter Cell Line for Tracking Human iPSC-Derived Cortical Interneurons"

_cells, 2022, doi:10.3390/cells11050853_

Round 1

Reviewer 1 Report

This is an excellent resource paper describing the generation of hPSC-based reporter lines for subtype-specific interneurons expressing LHX6.

Technically very good and process well described. Validation of reproter experssion is performed by in vitro differentiation using 2 protocols and show good expression of reporter in the directed protocol but not the targeted protocol.

I have the following changes to suggest:

  • make it clear in the section in the introduction relting to interneuron development what studies are from mouse and what are from human? This is relevant since the reporter lines will form human interneurons and most references are from mouse development.
  • Change last sentence in introduction “These LHX6 reporter lines faithfully express mCherry in LHX6+ neurons of hPSC-derived MGE progeny” to include both reporters.
  • The construct includes a p2A sequence but the correct cleavage and continued production of full size LHX is not confirmed.

The conclusion about mEmerald being better than mcherry as reporter is a correct interpretation of the data in this study but can not be a general claim and should be adjusted accordingly in the text.

Author Response

We appreciate very much the helpful suggestions by this reviewed that helped the improvement of this paper.

  • make it clear in the section in the introduction relating to interneuron development what studies are from mouse and what are from human? This is relevant since the reporter lines will form human interneurons and most references are from mouse development.

To help to distinguish mouse and human while maintaining the flow and conciseness of the text, we now refer the mouse proteins with one capital letter and all capital letters for human proteins (eg. Lhx6 for mouse and LHX6 for human).

We also moved one citation forward (Shi et al., Science, 2021, single cell RNAseq of human GE) as additional support for using LHX6 as reporter for human MGE-derived cortical interneurons.

  • Change last sentence in introduction “These LHX6 reporter lines faithfully express mCherry in LHX6+ neurons of hPSC-derived MGE progeny” to include both reporters.

Corrected as suggested.

  • The construct includes a p2A sequence but the correct cleavage and continued production of full size LHX is not confirmed.

The following observations support the production of functional LHX6 by our lines. 1) Both mCherry reporter lines and one of the mEmerald reporter line used in this study are LHX6 homozygous. We were able to detect LHX6 by antibody staining in almost all mCherry+ cells (Fig 3E, Fig S3) suggest production of decent levels of LHX6 protein. 2) LHX6 is required for cortical interneuron development, one would predict a reduced efficiency in the yield of reporter+ cells and GAD+ neurons in LHX6 homozygous lines in comparison to LHX6 heterozygous or WT lines. The comparable yield of GABAergic neurons by the homozygous lines to control PSC lines and similar number of mEmerald+ cells observed between the homozygous and heterozygous mEmerald lines indicate normal function of LHX6. We fully recognise however a Western blot analysis would be a better confirmation of the reviewer’s query if the revision time were not limited.

  • The conclusion about mEmerald being better than mcherry as reporter is a correct interpretation of the data in this study but can not be a general claim and should be adjusted accordingly in the text.

We thank the reviewer for this important point. The relevant text is reworded.

Reviewer 2 Report

In the manuscript "A novel iPSC Reporter Cell Line for Tracking Human Interneuron Differentiation and Modeling Disorders with Interneuronopathies", Drs. Maria Cruz Santos et al describe the establishment of a fluorescent reporter knock-in iPS cell line for the human LHX6 gene to monitor the efficiency of the generation of one of the human subtype-specific inhibitory neurons as a newly tool for the development of differentiation methods and pathological analysis.

LHX6 is particularly suitable as a gene to monitor MGE-derived cells among inhibitory neurons, and this reporter cell line should be approved as having a value. However, there are some important issues as a publication of a scientific paper, and it is unlikely that the quality of the paper will be high enough to be improved with some minor revisions to the content. Therefore, we recommend that authors change the content substantially or apply to a different journal.

Major issues

1: This manuscript describes a new generation of human LHX6 fluorescent reporter cells, which may imply that it is a report of a new resource. However, the authors themselves have already reported very similar result in Neuronal Signaling (2021) https://doi.org/10.1042/NS20210020. Therefore, this manuscript can never be accepted as a novel paper. It is true that LHX6-mEmerald itself may be a novel report, but LHX6-mCherry has already been reported. These are almost a synonym as a LHX6-reporter, although the type of fluorescence is different. Thus, it is not possible to publish a paper with the main purpose of establishing a new reporter line.

2: Although it is mentioned here to suggest that the fluorescence-positive cells are MGE-derived cells, it must also be sufficiently shown that this reporter line is different from that of CGE- or LGE-derived cells. A precise definition of MGE in the introduction should be followed by a comparative study of gene expression levels with those of CGE, LGE, and several more marker genes derived from them.

3: Although most of the immunostaining photographs are presented in a straightforward manner as merge photographs, it is necessary to present color-by-color photographs, including comparisons with DAPI/Hoechst, especially for the expression of LHX6 and mCherry, SST and mCheryy, and SST and GFP in Fig. 3.

4: The title gives the impression that the authors are describing disease modeling, but this expression is not appropriate because it is not actually done.

5: Regarding the picture of chromosomes in Fig2C, it is necessary to redo the G-band staining to show whether the chromosomes maintain their correct shape rather than the number of chromosomes per cell.

6: When the immunostaining images of Brachyury and EOMES in Fig2D are verified in mEmerald-KI cells, there is a possibility that the recognition wavelength of the reporter fluorescence may overlap. The results of these immunostaining experiments should be shown in a red color.

7: In Fig2B, staining for undifferentiated markers should be confirmed not only with transcription factors but also with membrane protein markers (e.g., using SSEA3, SSEA4, Tra1-81, Tr1-60, etc.).

8: In Fig3G, the mEmerald+ cells look good, but their association and interaction with surrounding cells should be discussed, so DAPI/Hoechst images or bright field images should be shown at the same time.

9: In Fig3DF, the fluorescence signal after d25 is shown, but it should be described how much the fluorescence signal appears in the short incubation period of d10 to d20.

10: In Fig3E, the merge rates of mCherry and LHX6, and mEmelard (or GFP) and LHX6 should be quantitatively evaluated.

11: In Fig4, it is interesting that the authors tried the SHIFTED method, but it would be more meaningful as a patterning experiment if the authors also show the results when SHH and PM are continuously added from d3 to d20.

12: For Fig. 4, it should be shown whether LHX6+ cells remain proliferative or not, and how they differ between STD and SHIFTED cells, which may have already been shown in the authors' previous paper.

13: In Fig4E, it makes sense to show the positive rate of SST. On the other hand, it should be better to show PV positivity (even if it is close to 0%) and subtype markers of inhibitory neurons other than MGE-derived. (for example, VIP, CALB, CALB2, CCK, NPY, etc.)

14: The results of demecolcine treatment are interesting, but this needs data with immunostaining and brightfield depiction so that you can see the cytotoxicity. It is also important to combine this with staining for proliferative markers, SST and other subtype markers.

15: In Fig. 4EF, GAD67 positive cells are mentioned, but it is also important to quantitatively analyze the positivity of excitatory neuronal markers such as CAMK2 and VGLUT1.

Author Response

1: This manuscript describes a new generation of human LHX6 fluorescent reporter cells, which may imply that it is a report of a new resource. However, the authors themselves have already reported very similar result in Neuronal Signaling (2021) https://doi.org/10.1042/NS20210020. Therefore, this manuscript can never be accepted as a novel paper. It is true that LHX6-mEmerald itself may be a novel report, but LHX6-mCherry has already been reported. These are almost a synonym as a LHX6-reporter, although the type of fluorescence is different. Thus, it is not possible to publish a paper with the main purpose of establishing a new reporter line.

The publication referred focused on the identification of TGFbeta signalling playing a role in cortical interneuron differentiation, which involves the use of the mCherry reporter line as a tool.  The current MS presents key information about the generation and prospective application of the reporter lines for researchers who may wish to obtain them for their own work or apply the same concept in designing bespoke tools of their own. We have already shared the mEmerald line with two laboratories outside the UK and have received additional expression of interest in this particular line. A dedicated publication on their establishment and properties would maximise the potential application of this valuable resource and to benefit  research of a broader community sooner.

2: Although it is mentioned here to suggest that the fluorescence-positive cells are MGE-derived cells, it must also be sufficiently shown that this reporter line is different from that of CGE- or LGE-derived cells. A precise definition of MGE in the introduction should be followed by a comparative study of gene expression levels with those of CGE, LGE, and several more marker genes derived from them.

We have added new text on MGE and CGE with respective key markers, as well as the developmental original of PV, SST and 5TH3aR interneurons. We also revised and added new text to highlight the restricted expression of LHX6 in the MGE, including a recent single cell transcriptomics analysis of the human GE (Shi et al, Science 2021). We believe these reference combined provides sufficient information to define LHX6+ cells as MGE-specific interneurons while minimising distraction from the main points.

3: Although most of the immunostaining photographs are presented in a straightforward manner as merge photographs, it is necessary to present color-by-color photographs, including comparisons with DAPI/Hoechst, especially for the expression of LHX6 and mCherry, SST and mCheryy, and SST and GFP in Fig. 3.

With the exception of live cell images in Fig 3G, color-by-color photographs of all immunostaining are now provided in revised figure 2B and new supplemental figures (figure S3-S5).

4: The title gives the impression that the authors are describing disease modeling, but this expression is not appropriate because it is not actually done.

We have removed the disease modelling part of the title. The revised title is ‘A novel LHX6 reporter line for tracking human iPSC-derived cortical interneurons’.

5: Regarding the picture of chromosomes in Fig2C, it is necessary to redo the G-band staining to show whether the chromosomes maintain their correct shape rather than the number of chromosomes per cell.

We attempted seeking expertise on G banding in the last two years as there isn’t necessary skills in house. However, due to Corvid pandemic and associated restrictions, the cytogenetics laboratories we have contacted stopped service to research projects in order to keep up with the health service.

However, as a routine practice, we perform microarray-based genetics analysis for all newly reprogramed iPSC lines and genetic manipulated hPSC lines. We use this analysis for verifying the sample identity and for identifying any newly acquired genetics changes (eg. copy number variance) by comparing data obtained from the source material or parental lines. This high stringent genetic integrity analysis provides more information than offered by the G banding. A summary report on all four reporter lines used in this study and their respective parental lines is provided as supplemental Table 1 in the revised paper.

6: When the immunostaining images of Brachyury and EOMES in Fig2D are verified in mEmerald-KI cells, there is a possibility that the recognition wavelength of the reporter fluorescence may overlap. The results of these immunostaining experiments should be shown in a red color.

This reviewer is right in considering this possibility. We make every effort where possible to minimise such possibility when performing immunocytochemistry. In this particular case, the excitation/emission range of the fluorescent tag (Alexa 647) of the 2nd antibody was sufficiently different from that of mEmerald. The green color in Fig 2D is pseudo-colored as is the case for many confocal images. Moreover, no signal at the mEmerald chanel was detected. This observation is consistent with a lack of mesoderm and endoderm expression of Lhx6 in the early developing embryo. (Grigoriou, M., et al, Development, 1998).

7: In Fig2B, staining for undifferentiated markers should be confirmed not only with transcription factors but also with membrane protein markers (e.g., using SSEA3, SSEA4, Tra1-81, Tr1-60, etc.).

A antibody staining for TRA 1-60 and TRA 1-82 was performed on both the mCherry and mEmerald lines and the new data is presented in the revised figure 2.

8: In Fig3G, the mEmerald+ cells look good, but their association and interaction with surrounding cells should be discussed, so DAPI/Hoechst images or bright field images should be shown at the same time.

Fig3G shows live cells. These images were deliberately taken in the dark field to better reveal the neuronal morphology. We however have shown mEmberald+ cells in bright fields in Fig 3F and Fig 4D.

9: In Fig3DF, the fluorescence signal after d25 is shown, but it should be described how much the fluorescence signal appears in the short incubation period of d10 to d20.

We have now added images of d5, d10 and d15 in figure S4. No mEmerald+ cells were detected in this time window. This observation is consistent with the finding in figure 3B-C, that the vast majority of day 21 cultures were NKX2.1+ progenitor cells.

10: In Fig3E, the merge rates of mCherry and LHX6, and mEmelard (or GFP) and LHX6 should be quantitatively evaluated.

The quantitative data is now included in Fig S3.

11: In Fig4, it is interesting that the authors tried the SHIFTED method, but it would be more meaningful as a patterning experiment if the authors also show the results when SHH and PM are continuously added from d3 to d20.

The aim of Fig4 was to provide evidence that the LHX6 reporter-expressing cells can increase or decrease in responding to culture conditions that affect interneuron production, either positive- or negatively.

By providing patterning cues before the peak of neural stem cell generation, the SHIFTED protocol is a suboptimal condition for interneuron induction designed to validate the reporter response to a potential negative regulator, while an example of positive response is provided in Fig 4G.

The experiment suggested by this reviewer might lead to an increased generation of NKX2.1+ MGE progenitors above the 90% already achieved in the STD protocol. NKX2.1+ MGE progenitors give rise to GABAergic interneurons (LHX6+) as well as cholinergic neurons (LHX8 lineage). It would be interesting to test whether a moderate change in NKX2.1+ proportion could translate to a change in LHX6+ cell numbers. This experiment is unfortunately outside of the scope of the 10 day revision time line.

12: For Fig. 4, it should be shown whether LHX6+ cells remain proliferative or not, and how they differ between STD and SHIFTED cells, which may have already been shown in the authors' previous paper.

There is abundant published literature demonstrating postmitotic expression of Lhx6/LHX6 in mice and man. Therefore, there doesn’t seem a reason to suspect the few LHX6+ cells produced in the SHIFTED condition would be different in nature compare to those in STD cultures.

13: In Fig4E, it makes sense to show the positive rate of SST. On the other hand, it should be better to show PV positivity (even if it is close to 0%) and subtype markers of inhibitory neurons other than MGE-derived. (for example, VIP, CALB, CALB2, CCK, NPY, etc.)

MGE gives rise to LHX6+ interneurons which later mature into either PV or SST subtypes. As indicated in the MS, no convincing PV+ cells were observed in these cultures within the time line analysed. Therefore, we believe that showing a reduction of SST+ cells under the SHIFTED condition is the appropriate way to demonstrate the point (that LHX6 reporter can respond to adverse interneuron differentiation cues).

14: The results of demecolcine treatment are interesting, but this needs data with immunostaining and brightfield depiction so that you can see the cytotoxicity. It is also important to combine this with staining for proliferative markers, SST and other subtype markers.

As part of the flow cytometry analysis, all samples were stained for DAPI for excluding the dead cells, this unbiased measurement revealed no difference in the proportion of cell death in cultures treated with or without demecolcine. Therefore, the observed increase in LHX6+ cells is unlikely due to cytotoxicity and preferential cell death. Moreover, if there were cytotoxicity, one might expect a relative decrease of LHX6+ population as postmitotic neurons are often more prone to cell death than neural progenitors.

We appreciate the reviewer’s point for further validation on cell proliferation and neuronal phenotype should time allow. The data presented as is however do provide sufficient evidence on the ability of the LHX6 reporter to respond to environmental cues.

15: In Fig. 4EF, GAD67 positive cells are mentioned, but it is also important to quantitatively analyze the positivity of excitatory neuronal markers such as CAMK2 and VGLUT1.

Similar to the response to comments 13 and 14, we appreciate that a detailed examination on the cellular identity in the SHIFTED condition (ie. a potential bias to generate cortical neurons) would be a bonus if time allows. It is our opinion however that the current data is adequate for demonstrating a responsiveness of the LHX6 reporter.

Reviewer 3 Report

Cruz Santos and coll. aim at introducing a novel iPSC reporter cell line for tracking human interneuron differentiation and modelling interneuropathies. The manuscript is informative it deals with a developing and innovative field, I therefore suggest to accept following major revision related mainly to clarify the methodology of the obtained data and conclusions. 

Major points: 

  1. the protocol related to cortical differentiation is lacking and no reference exists.
  2. In fig 3 B, the condition of the H7 Cortical is missing and the images are too small to be valuable. 
  3. In fig 3 E, Hoechst counterstaining is missing.
  4. In fig 3 I, the image is out of focus and aberrant colors are observed instead of green and red as indicated in the labe above. 
  5. In fig 4 D, the percentage of green cells at D35 appears less than at D25. 

Moderate English changes are required.

Author Response

We appreciate very much the helpful suggestions by this reviewer that helped the improvement of this paper.

  • the protocol related to cortical differentiation is lacking and no reference exists.

There was one sentence describing cortical differentiation in the original MS. Please refer 2.2 HESC culture and neuronal differentiation. ‘-----Cortical differentiation follows the same procedure without XAV, SHH and PM’.

In fig 3 B, the condition of the H7 Cortical is missing and the images are too small to be valuable.

The inclusion of cortical differentiation was to demonstrate a lack of LHX6-reporter expression under this condition. Since H7 control line doesn’t have a targeted reporter a side-by-side cortical differentiation control with this control line was not included in the design. H7 is however routinely used for cortical differentiation as part of other projects, we didn’t notice differences in behaviour between the LHM and H7 cells.

Fig 3B alone is now presented in a larger canvas in the supplemental file for inspection.

In fig 3 E, Hoechst counterstaining is missing.

DAPI counterstaining is now provided in Figure S3.

In fig 3 I, the image is out of focus and aberrant colors are observed instead of green and red as indicated in the labe above. 

We have provided an alternative image/field of view in the revised figure 3.

In fig 4 D, the percentage of green cells at D35 appears less than at D25. 

We appreciate the thorough examination by this reviewer. The overall proportion of mEmerald+ cells did increase from day 25 to day 35. There are however variations between field of views. The image for d35 was chosen as it nicely reveals neuronal morphology in an area of moderate density. It just happened that it wasn’t the best representation of the cultures as a whole.

Moderate English changes are required.

The revised MS has been proof read and edited by two native speakers.  We hope it is improved from the previous version.

Round 2

Reviewer 3 Report

The manuscript has been finely amended.